# Berry curvature contributions of kagome-lattice fragments in amorphous Fe–Sn thin films

Kohei Fujiwara [1,9] ✉, Yasuyuki Kato [2,9], Hitoshi Abe [3,4,5], Shun Noguchi[1], Junichi Shiogai[1,8], Yasuhiro Niwa [3,4], Hiroshi Kumigashira [3,6], Yukitoshi Motome [2] & Atsushi Tsukazaki [1,7]

Amorphous semiconductors are widely applied to electronic and energy-conversion devices owing to their high performance and simple fabrication processes. The topological concept of the Berry curvature is generally ill-defined in amorphous solids, due to the absence of long-range crystalline order. Here, we demonstrate that the Berry curvature in the short-range crystalline order of kagome-lattice fragments effectively contributes to the anomalous electrical and magneto-thermoelectric properties in Fe–Sn amorphous films. The Fe–Sn films on glass substrates exhibit large anomalous Hall and Nernst effects comparable to those of the single crystals of topological semimetals $Fe_3Sn_2$ and $Fe_3Sn$. With modelling, we reveal that the Berry curvature contribution in the amorphous state likely originates from randomly distributed kagome-lattice fragments. This microscopic interpretation sheds light on the topology of amorphous materials, which may lead to the realization of functional topological amorphous electronic devices.

In crystalline materials with long-range order of atoms/ions in the lattice, as depicted in Fig. 1a, the band dispersion defined in momentum ($k$) space is fundamental for interpreting their physical properties. The Berry curvature[1–4] determining the topological character of electronic bands is also formulated using $k$. In topological semimetals with linearly dispersed bands, such as magnetic Weyl semimetals and nodal line semimetals[4], the Berry curvature near the band singularities (Weyl points, nodal lines, etc.) leads to large anomalous Hall effect (AHE) and anomalous Nernst effect (ANE) (Fig. 1b)[5,6]. The intrinsic contributions derived from the Berry curvature are calculated theoretically via the following relations[2,5,6]:

$$\sigma_{AHE} = -\frac{e^2}{\hbar} \int [dk]\Theta(E - E_k)\Omega_z(k),  \tag{1}$$

$$\alpha_{xy} = -\frac{1}{e} \int dE \frac{\partial f}{\partial \mu} \sigma_{AHE}(E) \frac{E - \mu}{T},  \tag{2}$$

where $\sigma_{ANE}$ is anomalous Hall conductivity, $\alpha_{xy}$ anomalous Nernst conductivity, $e$ the elementary charge, $\hbar$ the reduced Planck constant, $E$ the energy, $\Theta$ the step function, $\Omega_z(k)$ the $z$-component of Berry curvature, $f$ the Fermi–Dirac function, $\mu$ the chemical potential, and $T$ temperature. The topological aspects of the electronic bands guarantee the giant electrical and magneto-thermoelectric responses, providing a reliable guideline for exploring new functional materials.

In the current framework based on the topological aspects, the understanding of amorphous materials without long-range order but

[1]Institute for Materials Research, Tohoku University, Sendai 980-8577, Japan. [2]Department of Applied Physics, University of Tokyo, Tokyo 113-8656, Japan. [3]Institute of Materials Structure Science, High Energy Accelerator Research Organization (KEK), Tsukuba 305-0801, Japan. [4]Department of Materials Structure Science, SOKENDAI (Graduate University of Advanced Studies), Tsukuba 305-0801, Japan. [5]Graduate School of Science and Engineering, Ibaraki University, Mito 310-8512, Japan. [6]Institute of Multidisciplinary Research for Advanced Materials, Tohoku University, Sendai 980-8577, Japan. [7]Center for Science and Innovation in Spintronics (CSIS), Core Research Cluster, Tohoku University, Sendai 980-8577, Japan. [8]Present address: Department of Physics, Osaka University, Toyonaka 560-0043, Japan. [9]These authors contributed equally: Kohei Fujiwara, Yasuyuki Kato. ✉e-mail: kohei.fujiwara@tohoku.ac.jp

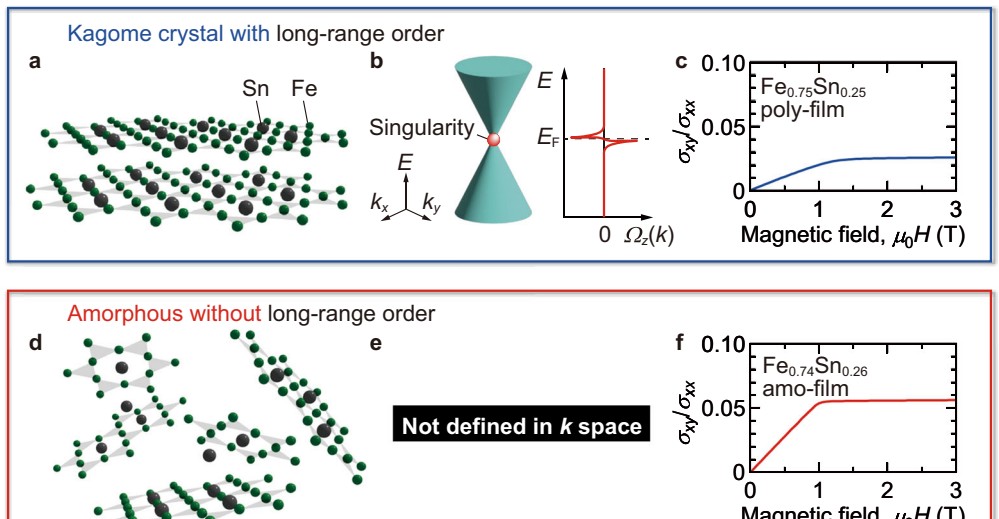

**Fig. 1 | General frameworks of kagome-lattice crystal and amorphous $Fe_xSn_{1-x}$.**
**a** Kagome-lattice crystal with long-range order in the lattice. The specific symmetry of the kagome-lattice[18,19] is discussed to contribute to the emergence of topological electronic states with **b** linearly dispersed bands in $k$ space. The $z$-component of Berry curvature $\Omega_z(k)$ becomes finite near the band singularity. When the Fermi energy $E_F$ is close to the $E$ of the band singularity, large intrinsic AHE and ANE are induced. **c** $\mu_0H$ dependences of the tangent of Hall angle $\sigma_{xy}/\sigma_{xx}$ for the $Fe_{0.75}Sn_{0.25}$ poly-film at $T = 300\,K$. **d** Amorphous without long-range order. The fragments with short-range kagome-lattice order, proposed as the microscopic picture of the $Fe_xSn_{1-x}$ amo-film in this study, are illustrated. **e** For such amorphous materials, $\Omega_z(k)$ is not defined. **f** $\mu_0H$ dependences of $\sigma_{xy}/\sigma_{xx}$ for the $Fe_{0.74}Sn_{0.26}$ amo-film at $T = 300\,K$.

with short-range order[7] (Fig. 1d), especially for topological materials, remains challenging[8–10], while the use of amorphous materials in various applications is expected owing to the low-cost and large-scale thin-film fabrication[6,11,12]. Without long-range order, Berry curvature is, in general, not defined explicitly using $k$ (Fig. 1e), because $k$ is no longer a good quantum number. Nevertheless, there have been some reports pointing out the observation of large AHE and ANE in amorphous and nanocrystalline films, for instance, ferromagnetic amorphous films of Fe–Si, Fe–Ge, Co–Si, Co–Ge, Fe–Co–Si (refs. 13,14), and Sm–Co (ref. 15) and nanocrystalline films of Fe–Sn (refs. 16,17). To interpret the large AHE in the Fe–Ge amorphous films[13], the authors suggested the dominant intrinsic contribution from the locally derived Berry curvature by calculating the energy-resolved density of Berry curvature using the density functional theory. However, this approach does not fully reflect the microscopic lattice feature of amorphous materials only with short-range order.

In this study, we discover large AHE and ANE in uniformly amorphous Fe–Sn films with no nanocrystalline domains, comparable to those of the single crystals of kagome-lattice topological semimetals $Fe_3Sn_2$ for AHE (ref. 18) and $Fe_3Sn$ for ANE (ref. 19), revealing that these effects are explained by the intrinsic mechanism based on Berry curvature in kagome-lattice fragments. Figure 1c, f shows the magnetic field $\mu_0H$ ($\mu_0$ being the vacuum permeability and $H$ the strength of out-of-plane magnetic field) dependences of tangent of Hall angle $\sigma_{xy}/\sigma_{xx}$ ($\sigma_{xy}$ being Hall conductivity and $\sigma_{xx}$ electrical conductivity) for an $Fe_{0.75}Sn_{0.25}$ polycrystalline film deposited at the substrate temperature $T_g = 400\,°C$ (poly-film; Supplementary Fig. 1) and an $Fe_{0.74}Sn_{0.26}$ amorphous film deposited at room temperature (amo-film), respectively. In addition to the sizable $\sigma_{xy}/\sigma_{xx}$ in the poly-film (Fig. 1c), the amo-film exhibits a significantly large $\sigma_{xy}/\sigma_{xx}$, comparable to those of topological ferromagnet crystals (Supplementary Fig. 2a). The $\sigma_{xy}/\sigma_{xx}$ of the poly-film may be suppressed by anisotropic grain formation leading only to small $\sigma_{xy}/\sigma_{xx}$. These facts motivate us to study the microscopic mechanism inducing the large AHE and ANE in the amo-films for boosting the exploration of giant responses in amorphous materials.

## Results

### Fe−Sn amorphous films without long-range order

We deposited the $Fe_xSn_{1-x}$ amo-films ($0.42 \leq x \leq 0.87$) on glass substrates at room temperature by co-sputtering ("Methods"). Figure 2a displays a typical transmission electron microscopy (TEM) image of the $Fe_{0.74}Sn_{0.26}$ film, showing no crystalline features, i.e., neither kagome-lattice layered structures[18,20] nor nanocrystalline domains as observed for $Fe_xSn_{1-x}$ films deposited on $Al_2O_3(0001)$ substrates[16]. A diffuse ring-like pattern in the selected area electron diffraction (Fig. 2a, inset) indicates no long-range order within the conventional TEM characterization. In the macroscopic X-ray diffraction (XRD) pattern shown in Fig. 2b, no diffraction peaks are discerned. This featureless XRD pattern is common to the other compositions (Supplementary Fig. 3a–f). Judging from these results, the room-temperature deposited $Fe_xSn_{1-x}$ films are categorized as an amorphous material in the examined $x$ range. Regarding the structural character of the $Fe_xSn_{1-x}$ amo-films, X-ray reflectivity measurement was employed to estimate the density $d$. An analysis based on a simple stack model (Fig. 2c) gives an excellent fit (red line) to the measured reflectivity (gray line), as shown in Fig. 2d (Supplementary Fig. 3g–l for the other $x$). The $x$ dependence of the $d$ is summarized in Fig. 2e (open red circles), together with the bulk reference data. The $d$ of the $Fe_xSn_{1-x}$ amo-films is almost comparable to the interpolated lines (solid gray lines) between $\beta$-Sn/$\alpha$-Sn and Fe expected for simple Fe−Sn alloys, and, more importantly, the values of FeSn, $Fe_3Sn_2$, and $Fe_3Sn$ bulks. Contrary to the intuitive sparse structure of amorphous, the Fe and Sn atoms densely form the $Fe_xSn_{1-x}$ amo-films.

### Large anomalous Hall and Nernst effects in Fe−Sn amorphous films

The characteristic features of topological electronic structure evidently appear in the electrical and magneto-thermoelectric properties. Taking the $Fe_{0.74}Sn_{0.26}$ amo-film as an example, we compare the magnetization $M$, Hall resistivity $\rho_{yx}$, and Nernst coefficient $S_{xy}$ versus $\mu_0H$ curves at $T = 300\,K$ in Fig. 3a–c. While the $M$–$\mu_0H$ curve of the amo-film shows no hysteresis due to in-plane magnetic anisotropy (Supplementary Fig. 4), the saturated $M$ value of $1.0 \times 10^6\,A\,m^{-1}$ is comparable to those of the $Fe_3Sn$ bulk crystal (cryst-bulk)[19,21] and the $Fe_3Sn$

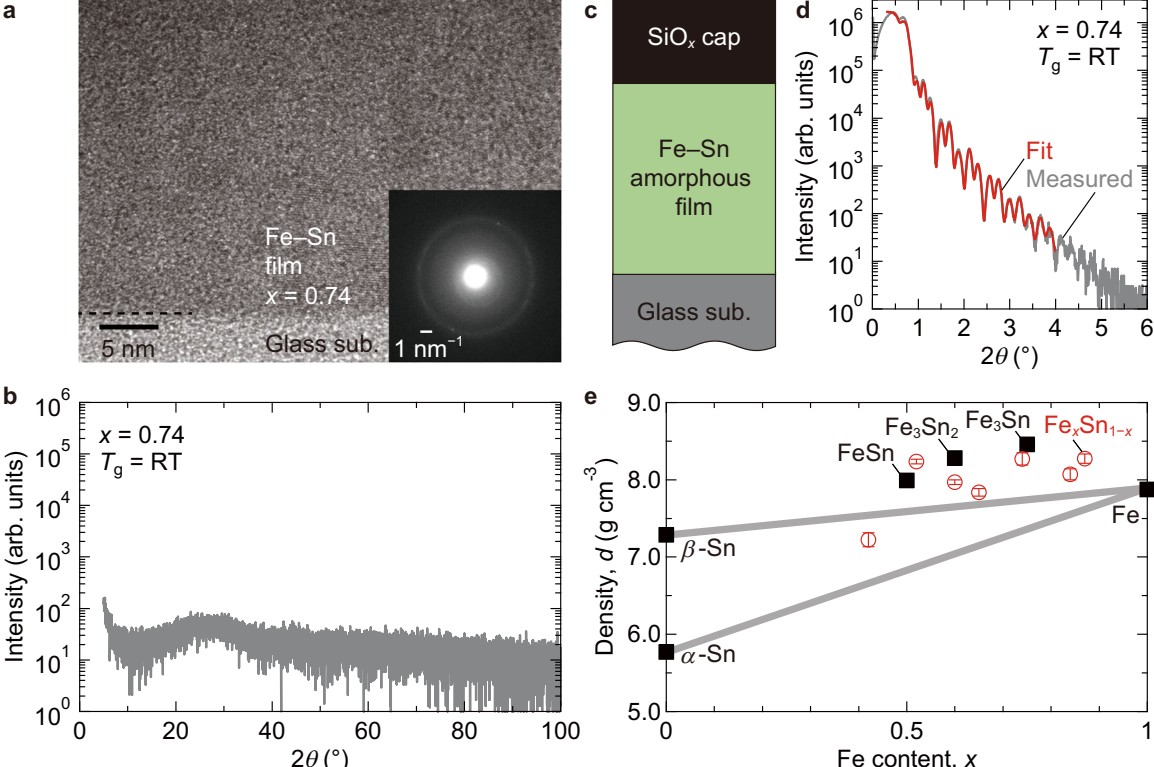

**Fig. 2 | Structural characterizations of $Fe_xSn_{1-x}$ amorphous films without long-range order. a** Cross-sectional TEM image of the $Fe_{0.74}Sn_{0.26}$ amo-film grown on glass at the substrate temperature ($T_g$) of room temperature (RT). The inset shows the selected area electron diffraction pattern. **b** Out-of-plane XRD pattern of the RT-grown $Fe_{0.74}Sn_{0.26}$ amo-film used for the TEM observation. The broad peak around 25° comes from the glass substrate. **c** The schematic structure of the film stack. **d** X-ray reflectivity data of the RT-grown $Fe_{0.74}Sn_{0.26}$ amo-film on glass. The gray and red curves are the measured data and the fitting result, respectively. **e** $x$ dependence of the $d$ estimated from the X-ray reflectivity data. For comparison, the bulk values in the database are included: JCPDS PDF No. 00-005-0390 for α-Sn, No. 00-004-0673 for β-Sn, No. 00-006-0696 for Fe, No. 01-071-8400 for FeSn, No. 01-071-0016 for $Fe_3Sn_2$, No. 01-074-5857 for $Fe_3Sn$. The two gray solid lines represent the interpolated lines between (Fe and α-Sn) and (Fe and β-Sn). The error bars represent the fitting errors.

crystalline film grown on $Pt/Al_2O_3(0001)$ (cryst-film)[20]. The $\rho_{yx}$–$\mu_0H$ and $S_{xy}$–$\mu_0H$ curves are consistent with the $M$–$\mu_0H$ curve, indicating AHE and ANE originated from the $z$-component of $M$, respectively. Figure 3d shows the $x$ dependence of the $\sigma_{AHE}/\sigma_{xx}$. The $\sigma_{AHE}/\sigma_{xx}$ (averaged over $\mu_0H$ = 2.5–3.0 T in the saturated state) of the $Fe_xSn_{1-x}$ amo-films, with the $\sigma_{xx}$ value consistent within the intrinsic region (Supplementary Fig. 2a), takes a broad maximum around $x$ = 0.75, as previously reported for the $Fe_xSn_{1-x}$ nanocrystalline films on $Al_2O_3(0001)$ substrates[16], which is comparable or even larger than those of the single crystals and crystalline films[18,19,22,23]. To quantify the magnitude of ANE, the $\alpha_{xy}$ is calculated using the relation of $\alpha_{xy} = \sigma_{AHE}S_{xx} + \sigma_{xx}S_{ANE}$, where $S_{xx}$ is Seebeck coefficient (Supplementary Fig. 5) and $S_{ANE}$ the anomalous component of $S_{xy}$; the $S_{ANE}$ is approximated by the $S_{xy}$ averaged for $\mu_0H$ = 2.5–3.0 T because the ordinary contribution is negligibly small (Fig. 3c). In Fig. 3e, the $\alpha_{xy}$ of the amo-films tends to increase with increasing $x$, reaching a large $\alpha_{xy}$ value of 1.3 A m$^{-1}$ K$^{-1}$ at $x$ = 0.87. Because such a large $\alpha_{xy}$ has so far been observed only in topological magnet crystals (Supplementary Fig. 2b), a mechanism distinct from extrinsic scattering should be invoked to explain the behavior in the amo-films. In addition, the small $\sigma_{AHE}/\sigma_{xx}$ and $\alpha_{xy}$ values of the Fe cryst-bulk[24] and cryst-films[25] point to the existence of their peaks in the Fe–Sn alloy compositions. The $\alpha_{xy}$ peak (Fig. 3e) and the $\sigma_{AHE}$ peak (Supplementary Fig. 5b) would appear at different $x$ values at $x$ > 0.9 and ~0.85, respectively, consistent with the relation of $\alpha_{xy} = \frac{\pi^2}{3}\frac{k_B^2 T}{e}\sigma'_{AHE}(E_F)$ (ref. 6) for the intrinsic ANE and AHE. Here, $\sigma'_{AHE}$ is the energy derivative of $\sigma_{AHE}$, $k_B$ is the Boltzmann constant, and the $E_F$ is the Fermi level; the appearance of $\alpha_{xy}$ and $\sigma_{AHE}$ peaks at different $E_F$ values is expected. Although varying $x$ not only shifts the $E_F$ but also modifies the electronic structure, these

tendencies satisfy one of the prerequisites for the intrinsic mechanism driven by Berry curvature[5,6]. In view of the potential use in ANE-type thermoelectric devices and thermal flow sensors[6,11,12], we also present the $S_{ANE}$ in Fig. 3f, which is a direct parameter evaluating the performance of thermoelectric conversion via ANE. The $S_{ANE}$ of 2.0 μV K$^{-1}$ for $x$ = 0.87 at $T$ = 300 K rivals the large values reported for the crystalline Fe-based binary alloys[19,25,26]. The facile synthesis of the $Fe_xSn_{1-x}$ amo-films by the sputtering method and the uniform amorphous texture, as well as the inexpensive and environmentally benign ingredients, will be great advantages for sustainable thermoelectric applications.

Following the scheme widely adopted to examine the validity of the Berry curvature-derived intrinsic AHE and ANE (refs. 5,6,27), we plot the $\alpha_{xy}$ against the $\sigma_{AHE}$ in Fig. 3g. Applying high-$T$ approximation to the two relations described in the introduction paragraph yields the ratio of $\alpha_{xy}/\sigma_{AHE}$ ~ $k_B/e$ (ref. 27). The $\alpha_{xy}/\sigma_{AHE}$ of the $Fe_xSn_{1-x}$ amo-films is as large as $k_B/10e$–$k_B/5e$ for the $\alpha_{xy}$ and $\sigma_{AHE}$ varying by roughly two orders of magnitude in the whole $x$ range. These $\alpha_{xy}/\sigma_{AHE}$ values are comparable to those of topological magnet crystals (Supplementary Fig. 2b). This systematic trend of $\alpha_{xy}/\sigma_{AHE}$ corroborates the intrinsic mechanism of the AHE and ANE in the $Fe_xSn_{1-x}$ amo-films.

**Short-range kagome-lattice order hidden in the amorphous structure**

To capture the microscopic arrangement of Fe atoms in the amo-films, we evaluated the Fe local environment by Fe $K$-edge X-ray absorption fine structure (XAFS) experiments. As depicted by the schematic model in Fig. 4a, extended XAFS (EXAFS) can sensitively probe the kinds of neighboring atoms around the absorbing element, here the Fe site, and the inter-atomic distances, which has been

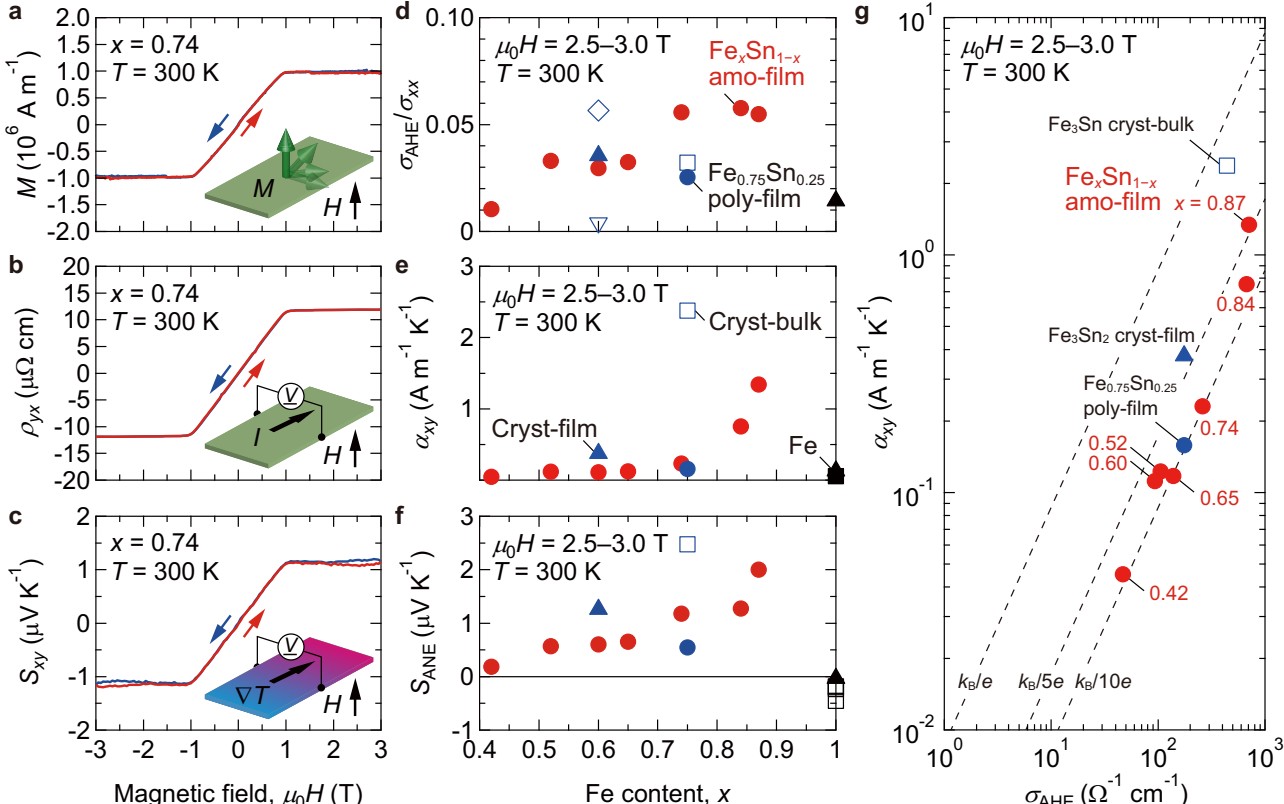

**Fig. 3 | Magnetic, electrical, and magneto-thermoelectric properties of $Fe_xSn_{1-x}$ amorphous films.** $\mu_0H$ dependences of **a** magnetization $M$, **b** Hall resistivity $\rho_{yx}$, and **c** Nernst coefficient $S_{xy}$ measured at $T = 300$ K for the $Fe_{0.74}Sn_{0.26}$ amo-film on glass. The insets show the schematic measurement configurations in an out-of-plane $\mu_0H$ ($V$: voltage, $I$: current). The blue and red curves correspond to the field-decreasing and -increasing scans. For the magneto-thermoelectric measurement, a temperature gradient of $(\nabla T)_x = 1.31$ K $mm^{-1}$. $x$ dependences of **d** tangent of Hall angle $\sigma_{AHE}/\sigma_{xx}$, **e** anomalous Nernst conductivity $\alpha_{xy}$, and **f** the anomalous component of Nernst coefficient $S_{ANE}$ for the $Fe_xSn_{1-x}$ amo-films (shown by the closed red circles) and the $Fe_{0.75}Sn_{0.25}$ poly-film (the closed blue circles). These data are obtained by averaging the measured electrical conductivity $\sigma_{xx}$, Hall conductivity $\sigma_{xy}$ for anomalous Hall conductivity $\sigma_{AHE}$, Seebeck coefficient $S_{xx}$, and $S_{xy}$ for $S_{ANE}$ between $\mu_0H = 2.5$–$3.0$ T in the saturated state. The error bars for these data, the standard deviations associated with the averaging, are smaller than the symbol size. For comparison, the data of the $Fe_3Sn_2$ cryst-bulks[18,22] (by the open blue diamond and triangle) and cryst-film[23] (the closed blue triangles), $Fe_3Sn$ cryst-bulk[19] (the open blue squares) and $Fe_3Sn$-containing poly-film (the closed blue circles), and Fe cryst-bulk[24] (the open black squares) and cryst-films[25] (the closed black triangles) are included. **g** $\alpha_{xy}$ versus $\sigma_{AHE}$ plot. The dashed lines from left to right represent the ratio of $k_B/e$, $k_B/5e$, and $k_B/10e$, respectively ($k_B$ is the Boltzmann constant and $e$ is the elementary charge). See Supplementary Fig. 2b for the detailed plot, including various topological ferromagnet crystals.

applied to determine the local coordination of amorphous oxides[28]. Figure 4b shows the Fourier-transformed EXAFS intensity (black curve) of the amo-film with $x = 0.74$ (see Supplementary Fig. 6 for the EXAFS spectra) and a $c$-axis-oriented $Fe_3Sn$ cryst-film on Pt/$Al_2O_3$(0001) (blue curve) as a crystalline film reference (Supplementary Fig. 1d). The strong peak intensities appear for the amo- and cryst-films at comparable radial distances of ~2 Å. For the spectral fitting to the amo-film data, we assume the elongation/shrinkage-associated deformation of $Fe_3Sn$ crystal with a conventional technique[29]. The basic fitting parameters are the degree of elongation/shrinkage and the effective coordination number. The red fitting curve by this model in Fig. 4b satisfactorily reproduces the amo-film data, with a reasonable degree of shrinkage of 2.2%. The fittings performed for the amo-films with $x = 0.52$ and 0.60 with respect to the FeSn and $Fe_3Sn_2$ crystals are also successful (Supplementary Fig. 7). These results are consistent with the concept based on the existence of short-range order that bears kagome-lattice-like local atomic configurations over the length scales comparable to the nearest Fe–Fe bonds (a few Å). In Fig. 4c, the $x$ dependence of the effective coordination number is plotted (filled red circles), which is normalized by the ideal values of the FeSn crystal for $x = 0.52$, the $Fe_3Sn_2$ crystal for $x = 0.60$, and the $Fe_3Sn$ crystal for $x = 0.74$. The normalized coordination number is smaller than unity, indicating

that the dense Fe and Sn atoms form short-range order in the amo-films, with imperfect connections as compared to the crystal lattice. The decrease of the normalized coordination number with increasing $x$ might come from the unstable high-$T$ phases of $Fe_3Sn_2$ and $Fe_3Sn$, while FeSn is stable over a wide $T$ range, including 300 K (ref. 30).

## Discussion

How small size kagome-like fragments are enough to produce the Berry curvature? Here we consider the short-range order on the length scale of ~ a few Å to exemplify the existence of kagome-lattice fragments in the amo-films. We conceived a theoretical model for the amorphous condition using a collection of kagome-lattice fragments that are compatible with the EXAFS results, as displayed in Fig. 4d, e. We construct a cube with $N \times N \times N$ cells of randomly oriented kagome-lattice fragments, which are defined with edge length $l$ in the unit of the Fe–Fe bond length in the kagome lattice. It should be stressed that this picture of amorphous on a sufficiently small length scale is distinct from the conventional polycrystal composed of crystallized domains with long-range order. Figure 4d displays an example for the amo-film with $x = 0.74$, which is expressed by $3 \times 3 \times 3$ cells of kagome-lattice fragments with $l = 4$. As shown in Fig. 4c, the calculated coordination number of the model (filled green circles) decreases with decreasing $l$,

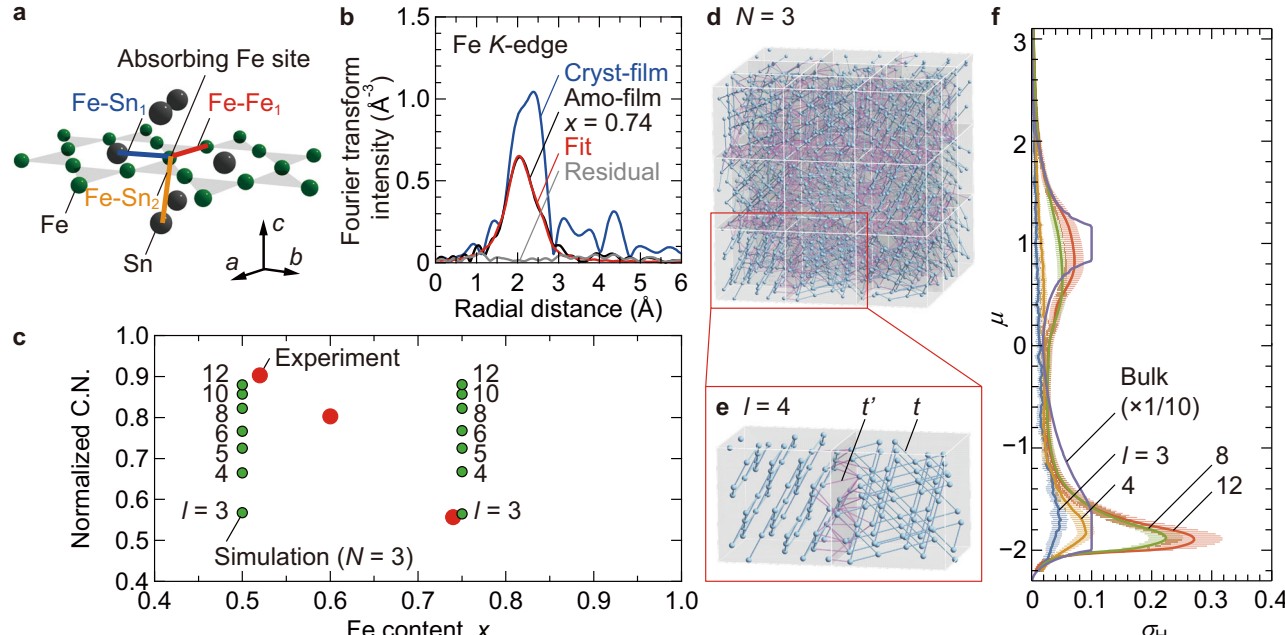

**Fig. 4 | Berry curvature contribution driven by short-range order of nano-sized kagome-lattice fragments. a** A structural model example of the $Fe_{0.52}Sn_{0.48}$ amo-film used for the EXAFS analysis. The $a$, $b$, and $c$ represent the crystallographic axes. By considering scattering contributions between the absorbing Fe site and primary neighboring sites, denoted as $Fe–Fe_1$, $Fe–Sn_1$, and $Fe–Sn_2$, the spectral fitting is performed. **b** Fourier-transformed Fe $K$-edge EXAFS intensities of the $Fe_{0.74}Sn_{0.26}$ amo-film on glass (shown by the black curves) and the $c$-axis oriented $Fe_3Sn$ cryst-film on $Pt/Al_2O_3(0001)$ (the blue curves). The red and gray curves show the fitting result to the amorphous data and the residual signal, respectively. **c** $x$ dependence of the normalized coordination number C.N. obtained by the EXAFS analysis and

calculated for the simulation models (**d**, **e**). **d** Kagome-lattice fragment model with a number of cubes per edge $N = 3$ and an edge length $l = 4$ used for the simulation of the $Fe_{0.74}Sn_{0.26}$ amo-film. **e** $Fe_3Sn$-like kagome-lattice fragments with $l = 4$. The $t$ and $t'$ are the hopping integrals between the nearest site pairs in each kagome plane (blue lines) and between the other site pairs (red lines), respectively. **f** Intrinsic Hall conductivity $\sigma_H$ calculated for $l = 3, 4, 8$, and 12 and the $Fe_3Sn$ bulk using the standard linear-response theory. The chemical potential $\mu$ is defined in the unit of the hopping integral $t = 1$. The error range represents the standard deviations obtained by the ten random arrangements of kagome-lattice fragments in the simulation.

indicating that the model from $l = 3$ to 12 captures the realistic amorphous condition by the bond disconnections at the cell boundaries. Using the standard linear-response theory ("Methods"), we calculated the $\mu$ dependence of the intrinsic Hall conductivity of the model structure, $\sigma_H$, for $l = 3, 4, 8$, and 12 and the $Fe_3Sn$ bulk. The $\sigma_H$ of $Fe_3Sn$ bulk in Fig. 4f shows saturating behavior at certain $\mu$, corresponding to the quantization. Noticeably, the kagome-lattice fragments even for $l = 3$ and 4, approximately a-few-nm scale, exhibit reasonable $\sigma_H$ values in the identical $\mu$ range, manifesting that the intrinsic contribution persists in the short-range kagome-lattice order. Given the close relationship between $\sigma_{AHE}$ and $\alpha_{xy}$ via Berry curvature, the observed large $\alpha_{xy}$ can presumably be interpreted by the identical model. This microscopic interpretation provides a significant step towards bridging the topological aspects and amorphous materials. The demonstrated scheme to evaluate theoretically the topological properties will contribute to the formalism of amorphous topological materials and the exploration of giant responses applicable to innovative amorphous devices.

## Methods
### Film growth
The samples were grown by radio-frequency magnetron co-sputtering[16] at an Ar gas pressure of 0.5 Pa and a radio-frequency power of 50 W. The surface of all the Fe–Sn films used in this study was capped with an ~15 nm-thick insulating $SiO_x$ layer to prevent oxidation. For the $Fe_xSn_{1-x}$ amo-films, Fe–Sn and $SiO_x$ layers were deposited on glass substrates (Matsunami Glass Ind., Ltd. S1126) at room temperature. The Fe contents of $x = 0.42–0.87$ were prepared by adjusting the target composition[16]. The thicknesses of these amo-films were 28–46 nm. For the $Fe_{0.75}Sn_{0.25}$ poly-film, 43 nm-thick Fe–Sn and $SiO_x$ layers were deposited on an $Al_2O_3(0001)$ substrate at $T_g = 400$ and

100 °C, respectively. For the FeSn, $Fe_3Sn_2$, and $Fe_3Sn$ (ref. 20) cryst-films, 4 nm-thick Pt, ~25 nm-thick Fe–Sn, and $SiO_x$ layers were deposited on $Al_2O_3(0001)$ substrates at $T_g = 600, 400$, and 100 °C, respectively. These bilayer and trilayer samples were fabricated without breaking the vacuum. Cross-sectional TEM and XRD using Cu $K_\alpha$ radiation were performed for the structural characterizations. Energy-dispersive X-ray spectroscopy was used to evaluate the composition of the films.

### Magnetization and transport measurements
The samples used for the transport measurements were identical to those for the structural characterizations. The $M$–$\mu_0H$ curves were measured with a vibrating sample magnetometer unit of a VersaLab (Quantum Design, Inc.) upon decreasing $\mu_0H$ from 3 T to −3 T and increasing $\mu_0H$ from −3 T to 3 T. By subtracting diamagnetic contributions from the substrate estimated by a linear fit to the data at $\mu_0H = 2–3$ T, the magnetization of the film was calculated. By anti-symmetrizing the decreasing-field and increasing-field data, the two anti-symmetrized $M$–$\mu_0H$ curves shown in Fig. 3a were obtained. For the electrical and thermoelectric transport measurements, the Fe–Sn film was patterned by photolithography and Ar-ion milling. The transport measurements were performed in the VersaLab using a home-made sample holder, in which a temperature gradient is generated by Joule heat from a resistor, and the temperature was monitored with on-chip resistance thermometers made of a sputtered Pt/Ti bilayer film. The transverse voltage induced by AHE and ANE was anti-symmetrized against $\mu_0H$ to eliminate spurious contributions arising from the misalignment of potential probes. The $S_{xx}$ contribution from the wiring components (gold wire and indium solder) was corrected. The plotted $T$ in the figures is the system temperature of the VersaLab.

## XAFS measurements

The Fe $K$-edge XAFS spectra were measured by the fluorescence yield mode at room temperature using a Lytle detector at the KEK Photon Factory beamline BL-9A. The EXAFS data were analyzed using the software ATHENA and ARTEMIS[31], and the FEFF6 code[32] was used to calculate theoretical EXAFS paths. The amplitude reduction (intrinsic loss) factor, $S_0^2$, was determined to be 0.752 by analyzing a standard 4-μm-thick Fe bulk foil sample because there are no standard Fe–Sn amorphous bulk samples. The $S_0^2$ was fixed throughout the EXAFS fittings to compare and discuss the coordination numbers among Fe–Sn amo-film samples (Supplementary Table 1). The spectral fittings to the Fourier transforms of the EXAFS spectra (Fig. 4b and Supplementary Figs. S6 and 7) were performed in the wave number $k$ and radial distance $r$ ranges of 2.4–13.1 Å$^{-1}$ and 1.5–2.9 Å for $x = 0.52$, 2.3–13.2 Å$^{-1}$ and 1.5–2.9 Å for $x = 0.60$, and 2.4–13.2 Å$^{-1}$ and 1.45–2.9 Å for $x = 0.74$.

## Simulation

We considered a model composed of a collection of nano-sized fragments of the ferromagnetic kagome-lattice material, Fe$_3$Sn. First, we considered the crystal structure assuming Fe$_3$Sn with uniformly randomly tilted kagome planes in a cubic block with edge length $l$. The translational degrees of freedom were also fixed uniformly randomly. We took the distance between the nearest neighbor sites of the kagome lattice as the unit of length, and then set $l$ to 3–12, which corresponds to a few nm. The stacking of the kagome layer is identical to that of Fe$_3$Sn while the interlayer distance is approximated to be $\sqrt{2/3}$. $N \times N \times N$ of the cubic blocks with independent random orientation of the nano-sized kagome fragments are arranged and stuck together with the periodic boundary condition (Fig. 4d). We set the hopping integral between the nearest site pairs in each kagome plane with distance 1 as $t = 1$, and those between the other site pairs with distances 1.4 or less as $t' = 0.5$ for simplicity (Fig. 4e). In addition to the hopping integrals, we considered the Kane-Mele-type spin–orbit coupling with $\lambda = 0.05$ for the $t = 1$ bonds as in ref. [18]. Assuming a ferromagnetically ordered state with magnetization parallel to the $z$ direction, we considered a spinless tight-binding Hamiltonian,

$$H = \sum_{\langle i,j \rangle} \left[ (t + i\lambda) c_i^\dagger c_j + \text{h.c.} \right] + \sum_{(i,j)} \left( t' c_i^\dagger c_j + \text{h.c.} \right), \qquad (3)$$

where the sums of $\langle i,j \rangle$ and $(i,j)$ run over all the $t$ bonds and all the $t'$ bonds, respectively. For the model with the kagome-lattice fragments as the small unit of kagome crystal, we computed the $\sigma_H$ in the plane perpendicular to the magnetization ($xy$-plane) by the standard Kubo formula, which reflects the peculiar Berry curvature of the kagome crystal. We generated ten different random structures independently for each $l$, and computed the mean and standard deviation of $\sigma_H$. The bold solid lines in Fig. 4f show the mean values of $\sigma_H$, and the ranges indicated by the thin bars show their standard deviations.

## Data availability

The data that support the findings of this study are available from the corresponding author upon request.

## Code availability

The simulation code is available from the corresponding author upon request.

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

## Acknowledgements

The authors thank S. Nishimura, T. Hirai, and K. Uchida for the development of the thermoelectric measurement system, S. Ito for the TEM analysis, T. Seki for helpful discussions, and NEOARK Corporation for the use of a maskless lithography system PALET. This work was performed under the GIMRT Program of the Institute for Materials Research, Tohoku University (Grant Nos. 202012-CRKEQ-0410 and 202112-CRKEQ-0413) and the approval of the Photon Factory Program Advisory Committee (Proposal Nos. 2021S2-002, 2021V006, and 2022G674). The numerical calculations were conducted on the supercomputer system at the Institute for Solid State Physics, University of Tokyo. This work was supported by JST CREST (JPMJCR18T2, A.T.) and the Thermal & Electric Energy Technology Foundation.

## Author contributions

K.F. grew the films and characterized their structural, magnetic, electrical, and magneto-thermoelectric properties. S.N. and J.S. contributed to the preliminary characterization of the magneto-thermoelectric properties. K.F. and H.A. performed the EXAFS analysis with the help of Y.N. and H.K. Y.K. and Y.M. performed the simulation. K.F., Y.K., Y.M., and A.T. wrote the manuscript based on the discussions with the other authors. A.T. conceived the project.

## Competing interests

The authors declare no competing interests.
