## [Peer Review File · Nature Communications]

Berry curvature contributions of kagome-lattice fragments in amorphous Fe–Sn thin filmsREVIEWER COMMENTS

Reviewer #1 (Remarks to the Author):

This work provides evidence of finite Berry Curvature arising in amorphous Fe-Sn films, even though these lack long range electronic order. Furthermore, it is concluded that the Berry Curvature is contributing to the anomalous electrical and magneto-thermal properties in the measured samples. Due to the simple fabrication process, I am sure that these properties can be widely applied to devices, so overall I support the publication of this work in Nature Communications.

However, there is one topic worth to discuss further:

The measured magnetisation seems to show the exact same behaviour as a function of H as ρ_{xy} and S_{xy} (Fig 3). In antiferromagnetic Kagome compounds like Mn₃Ge, that are mentioned several times by the authors, it is pointed out that the tiny ferromagnetic magnetisation that is still present in these materials, shows a different behaviour in field (<https://doi.org/10.1103/PhysRevB.100.085111>), which supports the claim that this anomalous behaviour of transverse transport coefficients is not caused by internal magnetic fields.

Furthermore, the authors claim that, as a function of doping, the peak of α_{xy} and σ_{xy} appear at different levels.

First: I get why this points to Berry curvature as an origin, how would the behaviour be if the magnetisation is responsible for the anomalous effects (wouldn't α_{xy} be the derivative of σ_{xy} as well)?

Second: For me the data is showing a nearly monotonic rise as a function of doping. To estimate the peak, are the authors including the black data points of pure Fe? Do the latter represent amorphous films as well?

I think it would strengthen the claim to discuss this matter a bit further, since in general I believe in the Berry Curvature occurring in these kind of Kagome systems, which is established in literature. Great to see that this is possible in amorphous films as well!

Another minor correction:

The caption of Supplemental figure 4 is stating a panel 4g, which is not shown in the figure

Reviewer #2 (Remarks to the Author):

Manuscript No: NCOMMS-23-08387-T

Title: Berry curvature contributions of kagome-lattice fragments in amorphous Fe-Sn thin films

Authors: Kohei Fujiwara et al.

In this manuscript, the authors have grown amorphous, polycrystalline and crystalline Fe-Sn films by rf magnetron sputtering by changing the substrates and deposition conditions. They have obtained exhibit large anomalous Hall and Nernst effects in the amorphous films which are comparable to those of the single crystals of Fe₃Sn₂ and Fe₃Sn. Using EXAFS measurements and theoretical calculations, the authors have shown that the short-range order of kagome-lattice fragments effectively contributes to the anomalous electrical and magneto-thermal properties in Fe-Sn amorphous films.

Amorphous semiconductor films generally can be grown on substrates at room temperature and hence can be deposited on flexible and a wide variety of substrates. Thus achieving bulk/crystal like properties in amorphous films has direct technological implication. Hence the topic discussed in this manuscript is important and needs fast attention of the scientific community.

However, I need one clarification before I can suggest it for publication.

EXAFS analysis is very important for this manuscript since the main conclusions have been drawn on

the basis of the EXAFS results. However, it is not clear from the manuscript how the authors have carried out the EXAFS data fitting, more details are needed to be given either in the manuscript or in the supplementary document. Particularly how the amplitude reduction factor has been chosen, whether it was varied or kept fixed, what is the value of the Debye-Waller factor etc. are to be clearly stated since all these are factors are important in accurate determination of the coordination numbers.

Reviewer #3 (Remarks to the Author):

The authors of the present paper reported that the Fe-Sn amorphous films on glass substrates exhibit large anomalous Hall and Nernst effects comparable to those of the single crystals of topological semimetals Fe_3Sn_2 and Fe_3Sn , and the amo-film also exhibits a significantly large value of tangent of Hall angle oxy/oxx , comparable to those of topological ferromagnet crystals. They further proposed a microscopic mechanism inducing the large anomalous Hall and Nernst effects in the amo-films. They presented that the Fe-Sn amorphous films have no long-range order but the short-range order of kagome-lattice fragments. They conceived a theoretical model for the amorphous condition using a collection of kagome-lattice fragments, and calculated the intrinsic Hall conductivity of the model structure using the standard linear-response theory. They claimed that the Berry curvature in the short-range order of kagome-lattice fragments effectively contributes to the anomalous electrical and magneto-thermal properties in Fe-Sn amorphous films.

The issue addressed in this paper is very interesting. The experimental results are also convincing. But, in spite of this, I am not convinced that this paper is suitable for publication in Nature Communications.

My main concern about this paper is the lack of novelty to satisfy the high bar imposed by Nature Communications.

1. The reports of the large anomalous Hall and Nernst effects in the Fe-Sn amorphous films have already been carried out.
2. In general, the Berry curvature is defined explicitly using k . For the Fe-Sn amorphous films without long-range order, k is no longer a good quantum number. How to define the Berry curvature in the Fe-Sn amorphous films with the short-range order of kagome-lattice fragments? The question has not been discussed in this paper.
3. The results of theory simulation about the Berry curvature contribution driven by short-range order of nano-sized kagome-lattice fragments is not convincing.

Therefore, I cannot recommend this paper for publication in Nature Communications. It is more appropriate to be published in a more specialized journal.

Attachment (1)

Response Letter

Manuscript ID: NCOMMS-23-08387-T

Responses to Reviewer #1's comments:

Comment 1: *This work provides evidence of finite Berry Curvature arising in amorphous Fe-Sn films, even though these lack long range electronic order. Furthermore, it is concluded that the Berry Curvature is contributing to the anomalous electrical and magneto-thermal properties in the measured samples.*

Due to the simple fabrication process, I am sure that these properties can be widely applied to devices, so overall I support the publication of this work in Nature Communications.

Response 1: We very much thank the reviewer for providing us with constructive comments/suggestions that helped us improve the manuscript. Our point-by-point responses to those comments/suggestions are described in the following.

Comment 1-1: *However, there is one topic worth to discuss further: The measured magnetisation seems to show the exact same behaviour as a function of H as ρ_{xy} and S_{xy} (Fig 3). In antiferromagnetic Kagome compounds like Mn_3Ge , that are mentioned several times by the authors, it is pointed out that the tiny ferromagnetic magnetisation that is still present in these materials, shows a different behaviour in field (<https://doi.org/10.1103/PhysRevB.100.085111>), which supports the claim that this anomalous behaviour of transverse transport coefficients is not caused by internal magnetic fields.*

Response 1-1: Thank you for this comment. The sample condition of the Fe-Sn amorphous films (amo-films) is completely different from antiferromagnetic kagome-lattice compounds like Mn_3Ge that was pointed out by the reviewer. As for AHE and ANE in ferromagnetic materials including the Fe-Sn amo-films, the same responses of M , ρ_{yx} , and S_{xy} as a function of $\mu_0 H$ (Figs. 3a–c) are the well-known typical character. The contribution of magnetization M to AHE is seen in a well-established empirical relation: $\rho_{yx} = R_0 \mu_0 H + R_A M$, where R_0 is the ordinary Hall coefficient and R_A the anomalous Hall coefficient. While the first term represents the ordinary Hall effect by Lorentz force, the second term corresponds to the AHE. In the intrinsic AHE mechanism, R_A is considered to reflect the contribution of Berry curvature (ref. ²). In antiferromagnetic materials with a vanishingly small M , the $\mu_0 H$ dependence of ρ_{yx} could deviate from that of M due to the other field-dependent contributions, which is an interesting topic for antiferromagnetic materials as discussed in the suggested reference. On the other hand, in ferromagnetic materials, particularly metals and semimetals with high carrier concentrations,

the $\mu_0 H$ dependence of M primarily governs that of ρ_{yx} because $\rho_{yx} = R_0 \mu_0 H + R_A M \sim R_A M$. Please note that this does not mean the insignificant contribution of R_A in ferromagnetic materials. The substantial contribution of R_A due to the Berry curvature from specific electronic bands can be common to kagome-lattice antiferromagnets like Mn_3Ge and Mn_3Sn and kagome-lattice fragments in the Fe-Sn amo-films. A similar discussion can apply to ANE. As shown in Fig. 2a, the Fe-Sn films deposited on glass at room temperature are uniformly amorphous without the segregation of crystallized domains. The observed M comes from the uniformly distributed Fe-Sn amorphous regions composed of the kagome-lattice fragments. Hence, the data shown in Figs. 3a–c are consistent with our claim of the large intrinsic AHE and ANE.

Comment 1-2: *Furthermore, the authors claim that, as a function of doping, the peak of α_{xy} and σ_{xy} appear at different levels. First: I get why this points to Berry curvature as an origin, how would the behaviour be if the magnetisation is responsible for the anomalous effects (wouldn't α_{xy} be the derivative of σ_{xy} as well)? Second: For me the data is showing a nearly monotonic rise as a function of doping. To estimate the peak, are the authors including the black data points of pure Fe? Do the latter represent amorphous films as well? I think it would strengthen the claim to discuss this matter a bit further, since in general I believe in the Berry Curvature occurring in these kind of Kagome systems, which is established in literature. Great to see that this is possible in amorphous films as well!*

Response 1-2: Regarding the first point, we again refer to the relation in Response 1-1, $\rho_{yx} = R_0 \mu_0 H + R_A M \sim R_A M$. Assuming the dominant contribution of M to the AHE under a constant R_A , the AHE should be enhanced with increasing the Fe content x . However, the $\sigma_{\text{AHE}}/\sigma_{xx}$ (Fig. 3d), which is a measure of the magnitude of AHE, shows the characteristic peak near $x = 0.8$ with significantly smaller values at $x \sim 0.4$ and $x = 1.0$, indicating that the R_A , namely, the Berry curvature is responsible for the AHE. For the ANE, it is difficult to discuss the contribution of M because the magneto-thermoelectric effect is not formulated with M . The ANE requires the orthogonal relationship between the directions of M , thermal gradient, and generated transverse electric field, which is in fact satisfied in our measurement configuration, as can be seen for the consistent shapes of the $M-\mu_0 H$ and $S_{xy}-\mu_0 H$ curves (Figs. 3a and c). For discussing the intrinsic Berry curvature origin of the ANE, we mention the relation of $\alpha_{xy} \propto \sigma'_{\text{AHE}}(E_F)$ (ref. ⁶). As a natural consequence of the differential function, it is expected that α_{xy} and σ_{AHE} take the peaks at different E_F values in the intrinsic ANE and AHE. In the original manuscript, we intended to point out that the appearance of the α_{xy} and σ_{AHE} peaks at seemingly different x values (Fig. 3e and Supplementary Fig. 5b), at least, is consistent with this expected behavior. However, given the semimetallic nature of $\text{Fe}_x\text{Sn}_{1-x}$ amo-films, the compositional variation not only shifts the E_F but also influences the electronic structure due to changes in the contributing orbital components. To elaborate on this point, we have revised the corresponding descriptions.

Page 8, Line 17–Page 9, Line 4:

“Additionally, under the assumption of the Fermi energy E_F shift induced by varying x in the amo-films, the result that the α_{xy} peak (Fig. 3e) and the σ_{AHE} peak (Supplementary Fig. 5b) would appear at different x values at $x > 0.9$ and ~ 0.85 , respectively, may agree with the relation of $\alpha_{xy} = \frac{\pi^2 k_B^2 T}{3e} \sigma'_{xy}(E_F)$ (ref. 6), where σ'_{xy} is the energy derivative of σ_{xy} and k_B is the Boltzmann constant, supporting the intrinsic mechanism driven by Berry curvature^{5,6}.”

has been revised to

“The α_{xy} peak (Fig. 3e) and the σ_{AHE} peak (Supplementary Fig. 5b) would appear at different x values at $x > 0.9$ and ~ 0.85 , respectively, **consistent** with the relation of $\alpha_{xy} = \frac{\pi^2 k_B^2 T}{3e} \sigma'_{\text{AHE}}(E_F)$ (ref. 6) **for the intrinsic ANE and AHE. Here, σ'_{AHE} is the energy derivative of σ_{AHE} and k_B is the Boltzmann constant, and the appearance of α_{xy} and σ_{AHE} peaks at different E_F values is expected. Although varying x not only shifts the E_F but also modifies the electronic structure, these tendencies satisfy one of the prerequisites for the intrinsic mechanism driven by Berry curvature^{5,6}.”**

As for the second point, our discussion is based on the data including those of Fe cryst-bulk²⁴ and cryst-films²⁵ in the literature to see the overall trends of the compositional dependence (Figs. 3d–f). Unfortunately, it is not easy to fabricate amo-films of pure Fe metal because of the thermodynamically stable crystalline bcc phase. In addition, we were not able to prepare pure Fe amo-films with our magnetron sputtering system because the sputtering rate is extremely low ($\ll 1$ nm/min) due to the magnetic interference between the Fe sputtering target and the magnetron cathode. Our $\text{Fe}_x\text{Sn}_{1-x}$ amo-films show significantly higher $\sigma_{\text{AHE}}/\sigma_{xx}$ and α_{xy} than the Fe cryst-bulk and cryst-films. Another point that should be noted in relation to Response 1-1 is that the M of $\text{Fe}_x\text{Sn}_{1-x}$ amo-films is lower than that of Fe (1.7×10^6 A m⁻¹ taken from Crangle, J. & Goodman, G. M., The Magnetization of Pure Iron and Nickel, Proc. R. Soc. Ser. A **321**, 477–491). These facts highlight the critical role of the kagome-lattice short-ranger order in the Fe-Sn amorphous alloy and the resulting finite local Berry curvature in the large AHE and ANE. According to the reviewer suggestion, we have added a brief description about this point to remove the confusion.

Page 8, Line 15–17:

“**Additionally, the small $\sigma_{\text{AHE}}/\sigma_{xx}$ and α_{xy} values of the Fe cryst-bulk²⁴ and cryst-films²⁵ points to the existence of their peaks in the Fe-Sn alloy compositions.**”

has been added.

Comment 1-3:

Another minor correction: The caption of Supplemental figure 4 is stating a panel 4g, which is not shown in the figure.

Response 1-3: We are sorry but we were not able to find the suggested mistake in the caption of Supplementary Fig. 4, because Supplementary Fig. 4 contains a single figure. If needed, we will make appropriate corrections.

Attachment (2)

Response Letter

Manuscript ID: NCOMMS-23-08387-T

Responses to Reviewer #2's comments:

Comment 2: *In this manuscript, the authors have grown amorphous, polycrystalline and crystalline Fe-Sn films by rf magnetron sputtering by changing the substrates and deposition conditions. They have obtained exhibit large anomalous Hall and Nernst effects in the amorphous films which are comparable to those of the single crystals of Fe₃Sn₂ and Fe₃Sn. Using EXAFS measurements and theoretical calculations, the authors have shown that the short-range order of kagome-lattice fragments effectively contributes to the anomalous electrical and magneto-thermal properties in Fe-Sn amorphous films.*

Amorphous semiconductor films generally can be grown on substrates at room temperature and hence can be deposited on flexible and a wide variety of substrates. Thus achieving bulk/crystal like properties in amorphous films has direct technological implication. Hence the topic discussed in this manuscript is important and needs fast attention of the scientific community.

Response 2: We thank you very much for reviewing our manuscript. As the reviewer stated, incorporating the superior functionality of topological semimetals into amorphous films will significantly accelerate the applications and the search for new materials with a fresh perspective. Our responses to his/her comments on the EXAFS analysis are shown below.

Comment 2-1: *However, I need one clarification before I can suggest it for publication. EXAFS analysis is very important for this manuscript since the main conclusions have been drawn on the basis of the EXAFS results. However, it is not clear from the manuscript how the authors have carried out the EXAFS data fitting, more details are needed to be given either in the manuscript or in the supplementary document. Particularly how the amplitude reduction factor has been chosen, whether it was varied or kept fixed, what is the value of the Debye-Waller factor etc. are to be clearly stated since all these are factors are important in accurate determination of the coordination numbers.*

Response 2-1: We apologize for the lack of details regarding the EXAFS analysis and parameters in the original manuscript. We have provided the information in the revised text and supplementary table as described below.

First, we determined the amplitude reduction factor (S_0^2) to be 0.752 by analyzing a standard 4- μm -thick Fe bulk foil sample. As there are no standard Fe-Sn amorphous *bulk*

samples, we adopted this S_0^2 value for the analysis of the Fe-Sn amo-films and fixed it throughout the EXAFS fittings of the three Fe-Sn amo-films. Although there might be a finite difference between the estimated and absolute true parameters due to the unavailability of the direct S_0^2 value, it is valid for the quantitative comparison among our Fe-Sn amo-film samples. To elaborate on this point, we have added the following sentences in the Methods section.

Page 14, Line 16–Page 15, Line 2, Methods, XAFS measurements:

“The amplitude reduction (intrinsic loss) factor, S_0^2 , was determined to be 0.752 by analyzing a standard 4- μm -thick Fe bulk foil sample because there are no standard Fe-Sn amorphous bulk samples. The S_0^2 was fixed throughout the EXAFS fittings to compare and discuss the coordination numbers among Fe-Sn amo-film samples (Supplementary Tab. 1).”

has been added.

As the reviewer pointed out, the Debye-Waller factors (σ^2) could affect the estimation of the coordination number for EXAFS fittings. Therefore, we should be careful about the Debye-Waller factors. The Debye-Waller factors obtained for our Fe-Sn amo-film samples and other fitting parameters are shown below in Supplementary Tab. 1. The variation in the normalized coordination number (C.N.) is clearly significant compared to the σ^2 . These data further support the decrease in the normalized C.N. discussed in our manuscript. We have added the table in Supplementary information. Again, thank you for this helpful comment.

Supplementary Table 1 has been added.

Supplementary Table 1. EXAFS fitting parameters. The amplitude reduction factor S_0^2 , the distance d and Debye-Waller factor σ^2 for the nearest neighboring Fe-Fe and Fe-Sn bonds, and normalized C.N. are shown.

Sample	S_0^2 (*)	Nearest neighboring Fe-Fe		Nearest neighboring Fe-Sn		Normalized C.N.
		d (Å)	σ^2 (Å ²)	d (Å)	σ^2 (Å ²)	
Amo-film $x = 0.52$	0.752	2.55	0.0127	2.55	0.0331	0.903
Amo-film $x = 0.60$	0.752	2.51	0.00813	2.62	0.0194	0.803
Amo-film $x = 0.74$	0.752	2.43	0.00567	2.70	0.0195	0.556

*The S_0^2 value determined by measuring a standard 4- μm -thick Fe bulk foil sample was fixed for these fittings.

Attachment (3)

Response Letter

Manuscript ID: NCOMMS-23-08387-T

Responses to Reviewer #3's comments:

Comment 3: *The authors of the present paper reported that the Fe-Sn amorphous films on glass substrates exhibit large anomalous Hall and Nernst effects comparable to those of the single crystals of topological semimetals Fe₃Sn₂ and Fe₃Sn, and the amo-film also exhibits a significantly large value of tangent of Hall angle σ_{xy}/σ_{xx} , comparable to those of topological ferromagnet crystals. They further proposed a microscopic mechanism inducing the large anomalous Hall and Nernst effects in the amo-films. They presented that the Fe-Sn amorphous films have no long-range order but the short-range order of kagome-lattice fragments. They conceived a theoretical model for the amorphous condition using a collection of kagome-lattice fragments, and calculated the intrinsic Hall conductivity of the model structure using the standard linear-response theory. They claimed that the Berry curvature in the short-range order of kagome-lattice fragments effectively contributes to the anomalous electrical and magneto-thermal properties in Fe-Sn amorphous films.*

The issue addressed in this paper is very interesting. The experimental results are also convincing. But, in spite of this, I am not convinced that this paper is suitable for publication in Nature Communications.

My main concern about this paper is the lack of novelty to satisfy the high bar imposed by Nature Communications.

Response 3: We greatly appreciate the reviewer for reviewing our manuscript. In the following, we would like to address his/her concerns about the current work's novelty and the results of the simulation.

Comment 3-1: *1. The reports of the large anomalous Hall and Nernst effects in the Fe-Sn amorphous films have already been carried out.*

Response 3-1: To the best of our knowledge, there have so far been no reports on the large anomalous Hall effect (AHE) and anomalous Nernst effect (ANE) in Fe-Sn *amorphous* films. In our previous study¹⁶, we have reported AHE in Fe-Sn *nanocrystalline* films grown on Al₂O₃(0001). The nanocrystalline films contain kagome-lattice-like crystallized nanodomains detectable by conventional transmission electron microscopy, which are distinct from Fe-Sn amorphous films presented in this study. The uniformly amorphous condition, excluding the

contributions of the nanocrystalline domains, is critically important to disentangle the roles of the presence/absence of long-range order in the emergence of large AHE and ANE. In the current work, we suppressed the crystallization using glass substrates and room-temperature sputtering, revealing the electrical and magneto-thermoelectric properties of Fe-Sn amorphous films for the first time. The synthesis of the uniformly amorphous films and the new discovery of the large AHE and ANE emerging even without long-range order and nanocrystalline domains has led us to recognize the importance of Berry phase physics in the amorphous system. To solve the confusion related to this point and make clearer descriptions, we have revised the text as follows.

Page 5, Line 7:

“Nevertheless, there have been some reports pointing out the observation of large AHE and ANE in amorphous and/or nanocrystalline films, for instance, ferromagnetic amorphous films of Fe-Si, Fe-Ge, Co-Si, Co-Ge, Fe-Co-Si (refs. ^{13,14}), Sm-Co (ref. ¹⁵), and Fe-Sn (refs. ^{16,17}).”

has been revised to

“Nevertheless, there have been some reports pointing out the observation of large AHE and ANE in amorphous and/or nanocrystalline films, for instance, ferromagnetic amorphous films of Fe-Si, Fe-Ge, Co-Si, Co-Ge, Fe-Co-Si (refs. ^{13,14}), Sm-Co (ref. ¹⁵) and **nanocrystalline films of Fe-Sn** (refs. ^{16,17}).”

Page 5, Line 12–15:

“In this study, we reveal that the large AHE and ANE in the amorphous Fe-Sn films, which are comparable to those of the single crystals of kagome-lattice topological semimetals Fe₃Sn₂ for AHE (ref. ¹⁸) and Fe₃Sn for ANE (ref. ¹⁹), are explained by the intrinsic mechanism based on Berry curvature in kagome-lattice fragments.”

has been revised to

“In this study, we **discover** large AHE and ANE in **uniformly** amorphous Fe-Sn films with **no nanocrystalline domains**, comparable to those of the single crystals of kagome-lattice topological semimetals Fe₃Sn₂ for AHE (ref. ¹⁸) and Fe₃Sn for ANE (ref. ¹⁹), **revealing that these effects** are explained by the intrinsic mechanism based on Berry curvature in kagome-lattice fragments.”

Comment 3-2: 2. In general, the Berry curvature is defined explicitly using k . For the Fe-Sn amorphous films without long-range order, k is no longer a good quantum number. How to define the Berry curvature in the Fe-Sn amorphous films with the short-range order of kagome-lattice fragments? The question has not been discussed in this paper.

Response 3-2: Thank you for this comment. As explained in the Methods section for the simulation, we considered the Fe₃Sn-based crystal structure with randomly tilted kagome planes in a cubic block with an edge length l and fixed the translational degrees of freedom uniformly random. For this model with the kagome-lattice fragments as the small unit of kagome crystal, we computed the Hall conductivity σ_H in the plane perpendicular to the magnetization (xy -plane) by the standard Kubo formula, which reflects the peculiar Berry curvature of the kagome crystal. Although the Berry curvature is not well defined when the translation symmetry is broken as the reviewer pointed out, we believe that the origin of the AHE and ANE in our proposed kagome-lattice fragments is commonly understood by the Berry phase physics, since the σ_H changes systematically with increasing the fragment size and approaches the bulk behavior with translation symmetry, as shown in Fig. 4f (see also Response 3-3 below). We therefore use the term “Berry curvature contribution” to represent the common physics, despite the lack of mathematical definition. Please note that “Berry curvature” was similarly used in Refs. 13 and 14 for disordered systems where k is not a good quantum number.

Page 15, Line 18–Page 16, Line 7, in Methods for the simulation, we revised the descriptions. “Assuming a ferromagnetically ordered state with magnetization parallel to the z direction, we considered a spinless tight-binding Hamiltonian to calculate the σ_H in the plane perpendicular to the magnetization (xy -plane) is computed by the standard Kubo formula. The Hamiltonian reads

$$H = \sum_{\langle i,j \rangle} [(t + i\lambda)c_i^\dagger c_j + \text{h.c.}] + \sum_{(i,j)} (t' c_i^\dagger c_j + \text{h.c.}),$$

where the sums of $\langle i,j \rangle$ and (i,j) run over all the t bonds and all the t' bonds, respectively.”

has been revised to

“Assuming a ferromagnetically ordered state with magnetization parallel to the z direction, we considered a spinless tight-binding Hamiltonian,

$$H = \sum_{\langle i,j \rangle} [(t + i\lambda)c_i^\dagger c_j + \text{h.c.}] + \sum_{(i,j)} (t' c_i^\dagger c_j + \text{h.c.}),$$

where the sums of $\langle i,j \rangle$ and (i,j) run over all the t bonds and all the t' bonds, respectively.

For the model with the kagome-lattice fragments as the small unit of kagome crystal, we computed the σ_H in the plane perpendicular to the magnetization (xy -plane) by the standard Kubo formula, which reflects the peculiar Berry curvature of the kagome crystal.”

Comment 3-3: 3. The results of theory simulation about the Berry curvature contribution driven by short-range order of nano-sized kagome-lattice fragments is not convincing.

Response 3-3: We interpret the comment “not convincing” as casting a doubt on our calculation results and would like to explain why we believe they are reasonable. Our model is composed of a periodic array of N^3 kagome-lattice fragments, each of which has an edge length of l . Thus, it should reproduce the bulk behavior in the infinite limit of l . We carefully confirmed this by calculating (i) up to $l = 35$ with $N = 1$, (ii) up to $l = 12$ with $N = 3$ (Fig. 4f), and (iii) up to $l = 6$ with $N = 4$ (Figure R1 shown below). All these results approached the expected bulk behavior in a systematic manner. Considering the reviewer’s concern, we also performed the calculation for $N = 4$ with varying l , as shown in Figure R1. The saturating σ_H behavior of the bulk limit agrees with that for $N = 3$ (Fig. 4f) irrespective of the system size N used. Additionally, the sizable σ_H , consistent with the intrinsic origin, again emerges in small kagome-lattice fragments of $l = 3$ and 4. These results underpin the validity of our calculation results.

Figure R1. **a**, Kagome-lattice fragment model with $N = 4$ and $l = 4$ used for the simulation of the $\text{Fe}_{0.74}\text{Sn}_{0.26}$ amo-film. **b**, Fe_3Sn -like kagome-lattice fragments with $l = 4$. **c**, σ_H calculated for $l = 3, 4, 5$, and 6 and the Fe_3Sn bulk using the standard linear-response theory. The saturating σ_H behavior of the bulk agrees with that for $N = 3$ (Fig. 4f), showing that our model reproduces well the bulk behavior irrespective of the N value used. The μ is defined in the unit of the hopping integral $t = 1$. The error bars represent the standard deviations obtained by the ten random arrangements of kagome-lattice fragments in the simulation.

Comment 3-4: *Therefore, I cannot recommend this paper for publication in Nature Communications. It is more appropriate to be published in a more specialized journal.*

Response 3-4: As explained above, we believe that this manuscript provides general readers working in various fields of basic science and device engineering with the impressive experimental discovery and microscopic interpretation of the large AHE and ANE in the uniformly amorphous Fe-Sn films. Considering the positive evaluations from Reviewers #1 and #2, this work contains many important implications that can greatly advance the applications of topological materials. Our approach, which combines experimental characterizations with simulation, provides a clear direction for understanding the elusive Berry phase physics in amorphous materials. Therefore, we are confident that the current achievements meet the high standards for publication in *Nature Communications*.

Attachment (4)

List of Revisions

Manuscript ID: NCOMMS-23-08387-T

1. Page 5, Line 7: The sentence has been revised (Response 3-1).
2. Page 5, Line 12–15: The sentence has been revised (Response 3-1).
3. Page 8, Line 15–17: A new sentence has been added (Response 1-2).
4. Page 8, Line 17–Page 9, Line 4: The sentences have been revised (Response 1-2).
5. Page 14, Line 16–Page 15, Line 2: New sentences have been added (Response 2-1).
6. Page 15, Line 18–Page 16, Line 7: The previous sentence has been revised and a new sentence has been added (Response 3-2).
7. Supplementary information: Supplementary Table 1 has been added (Response 2-1).

In addition to these revisions to respond to the reviewers' comments, the following revisions have been made to improve the correctness of explanations and fix the errors in the original manuscript.

8. We have replaced the term “magneto-thermal” with “**magneto-thermoelectric**” (Page 3, Line 8, Page 4, Line 15, Page 7, Line 14, Page 17, Line 12–14, and Page 26, Line 1).
9. We have revised the expressions of variables as follows: anomalous Hall conductivity σ_{AHE} , the anomalous component of Nernst coefficient (S_{xy}) and **anomalous Nernst** conductivity α_{xy} . Associated with this, the vertical axis labels of Figs. 3d, f, Supplementary Fig. 2a, and Supplementary Fig. 3b and the horizontal axis labels of Fig. 3g and Supplementary Fig. 2b have been revised. The plotted values have not been changed from the original manuscript.
10. Linked to the above change, in Page 8, Line 9–11, the following sentence has been revised. “To quantify the magnitude of ANE, the α_{xy} is calculated using the relation of $\alpha_{xy} = \sigma_{\text{AHE}}S_{xx} + \sigma_{xx}S_{\text{ANE}}$, where S_{xx} is Seebeck coefficient (Supplementary Fig. 5) and **S_{ANE} the anomalous component of S_{xy} ; the S_{ANE} is approximated by the S_{xy} averaged for $\mu_0H = 2.5\text{--}3.0$ T because the ordinary contribution is negligibly small (Fig. 3c).**”
Also, the descriptions related to σ_{AHE} and S_{ANE} in the text and Supplementary Information have been revised.
11. Page 14, Line 11–12: A sentence about the definition of the plotted T , “**The plotted T in the figures is the system temperature of the VersaLab.**”, has been added in the Methods section.

12. The caption of Fig. 3 has been revised as follows:

“**Figure 3...** c , S_{xy} measured at $T = 300$ K for the $\text{Fe}_{0.74}\text{Sn}_{0.26}$ amo-film on glass. The insets show the schematic measurement configurations in an out-of-plane $\mu_0 H$. The blue and red curves correspond to the field-decreasing and -increasing scans. **For the magnetothermoelectric measurement, a temperature gradient of $(\nabla T)_x = 1.31$ K mm^{-1} .** x dependences of \mathbf{d} , $\sigma_{\text{AHE}}/\sigma_{xx}$, \mathbf{e} , α_{xy} , and \mathbf{f} , S_{ANE} for the $\text{Fe}_x\text{Sn}_{1-x}$ amo-films (shown by the closed red circles) and the $\text{Fe}_{0.75}\text{Sn}_{0.25}$ poly-film (the closed blue circles). These data are **obtained by averaging the measured σ_{xx} , σ_{xy} for σ_{AHE} , S_{xx} , and S_{xy} for S_{ANE} between $\mu_0 H = 2.5$ – 3.0 T in the saturated state...**”

13. The caption of Fig. 4 has been revised to fix the errors in the previous manuscript:

“**Figure 4...** \mathbf{f} , σ_{H} calculated for $l = 3, 4, 8,$ and 12 and the Fe_3Sn bulk using the standard linear-response theory. The μ is defined in the unit of the **hopping integral $t = 1$** . The error bars represent the standard deviations obtained by the ten **random arrangements of kagome-lattice fragments in the simulation.**”

These additional revisions do not affect any conclusion presented in the manuscript.

REVIEWERS' COMMENTS

Reviewer #1 (Remarks to the Author):

I am convinced by the authors explanations and furthermore agree to the changes of the manuscript. I am recommending this work for publication in Nature Communications. All the best for future research on this topic!

Reviewer #2 (Remarks to the Author):

I am satisfied with the modifications carried out by the authors in the revised manuscript and suggests its publication.

Reviewer #3 (Remarks to the Author):

The authors have successfully replied my comments and made some refinements to the presentation of the results in the revised version of the manuscript, thus improves the quality of the manuscript. I agree that this work is very impressive and contains many important advance in the applications of topological materials. Now, I recommend the paper for publication in Nature Communications.